# Transmission of heat modes across a potential barrier

Amir Rosenblatt[1], Fabien Lafont[1], Ivan Levkivskyi[2], Ron Sabo[1], Itamar Gurman[1], Daniel Banitt[1], Moty Heiblum[1] & Vladimir Umansky[1]

Controlling the transmission of electrical current using a quantum point contact constriction paved a way to a large variety of experiments in mesoscopic physics. The increasing interest in heat transfer in such systems fosters questions about possible manipulations of quantum heat modes that do not carry net charge (neutral modes). Here we study the transmission of upstream neutral modes through a quantum point contact in fractional hole-conjugate quantum Hall states. Employing two different measurement techniques, we were able to render the relative spatial distribution of these chargeless modes with their charged counterparts. In these states, which were found to harbor more than one downstream charge mode, the upstream neutral modes are found to flow with the inner charge mode—as theoretically predicted. These results unveil a universal upstream heat current structure and open the path for more complex engineering of heat flows and cooling mechanisms in quantum nano-electronic devices.

[1] Braun Center for Submicron Research, Dept. of Condensed Matter physics, Weizmann Institute of Science, Rehovot 76100, Israel. [2] Institute of Ecology and Evolution, University of Bern, CH-3012 Bern, Switzerland. Amir Rosenblatt and Fabien Lafont contributed equally to this work. Correspondence and requests for materials should be addressed to F.L. (email: lafont.fabien@gmail.com)

The intimate link between heat current, entropy flow and therefore information transfer[1,2], triggered recent interest in thermoelectric effects occurring at the nanoscale, such as measurements of the quantum limit of heat flow of a single quantum mode[3–6], heat Coulomb blockade of a ballistic channel[7] or quantum-limited efficiency of heat engines and refrigerators[8,9]. One main experimental obstacle in measuring thermal effects is to decouple the charge from heat currents. Such separation is made easier in the fractional quantum Hall effect (FQHE)[10], since, at least in hole-conjugate states (say, $1/2 < \nu < 1$), charge-less heat modes propagate with an opposite chirality (upstream) to that of the charge modes[11–13]. Theoretically, while Laughlin wave function[14] has a great success in describing various aspects of the FQHE at filling factors $\nu = 1/m$, with odd $m$, the structure of hole-conjugate filling fractions, such as $\nu = 2/3$ and $\nu = 3/5$, is still not clear. Two edge-model structures had been proposed for the most studied $\nu = 2/3$ state, the first considers this state as a charge conjugate of the $\nu = 1/3$ state; namely, a $\nu = 1/3$ hole Landau level (LL) in the completely filled $\nu = 1$ electronic LL form a Laughlin condensate with $\nu = 1/3$[15,16]. The second considers it as a $\nu = 1$ type condensate of fractional $e^{*} = 1/3e$ quasiparticles on top of the $\nu = 1/3$ state[17,18]. Interestingly, recent experiments with a softly defined edge potential (induced by a gate) showed that the structure of the edge charge modes at filling factor $\nu = 2/3$ (and similarly at $\nu = 3/5$) is in fact composed of two spatially separated charge modes[19], which would suggest the second point of view to be more appropriate. Furthermore, topological arguments require to have a conserved total number of net modes, therefore predicting two upstream neutral modes at $\nu = 2/3$, and three at $\nu = 3/5$.

Here we fully characterize the transmission of the downstream charge modes and the upstream neutral modes through a potential barrier, imposed by a quantum point contact (QPC). We show their relative spatial distribution and heat power carried by the neutral modes. Using the same device, we shed more light on the interplay between charge and neutral modes impinging on a QPC by distinguishing thermal fluctuations from shot noise.

## Results

**Transmission of neutral modes across a QPC.** The experimental setup, designed to map the transmission of neutral modes through a QPC constriction, is presented in Fig. 1a, c, e. The QPC (on the right) partitions either the charge mode emitted from S2

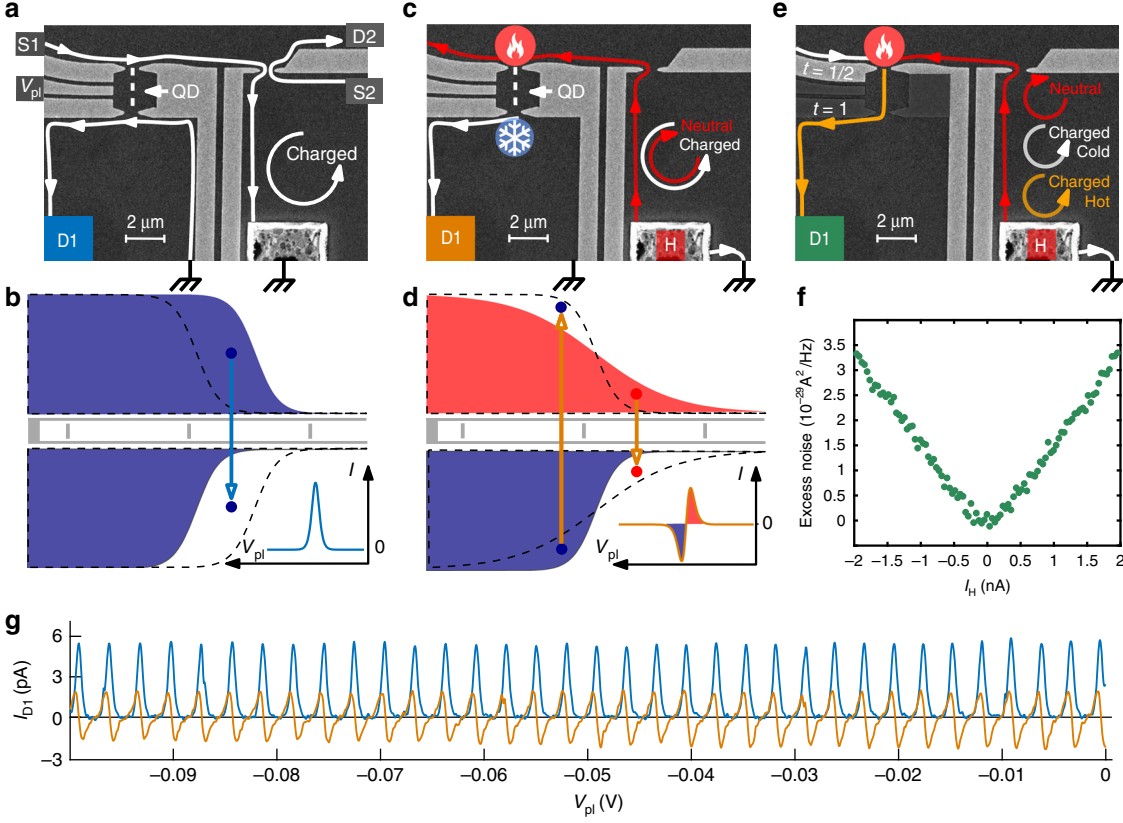

**Fig. 1** Description of the experimental device and methods. **a** Scanning electron microscope (SEM) image of the device. In this configuration, the current is sourced from S1, reaching to the quantum dot (QD). Sweeping the plunger gate voltage results in a succession of Coulomb peaks as shown by the blue curve on **g**. **b** Schematic of the equilibrium distribution on each side of the QD. When the plunger gate is tuned, electron can tunnel into the dot from the high occupation number region to the lower one, creating a positive current measured at D1. **c** Neutral mode heat detection configuration: The current sourced in H is directed to the ground and plays no part in the experiment. A hot spot present at the upstream side of contact H excites the neutral heat modes flowing upstream toward the QD, creating a thermal gradient across the QD. The produced thermoelectric current then flows to D1. **d** Sketch of the equilibrium distribution on both side of the dot in the case depicted in **c**. In this case, the tunneling direction will depend if an energy level of the QD is placed below or above the center of the distribution. This induces an alternating current when the plunger gate is tuned. **e** Noise measurement configuration: Here the input QPC of the QD is set to half transmission while the second one is fully open. This turns the QD into an effective single QPC device. The heat carried by the neutral modes increase the electron temperature at the input QPC of the dot, which increases the Johnson–Nyquist noise, measured at D1. **f** Excess noise measured at D1 as function of current injected in H, as described in **e**. **g** Measurement corresponding to the configuration **a** in blue and **c** in orange

 

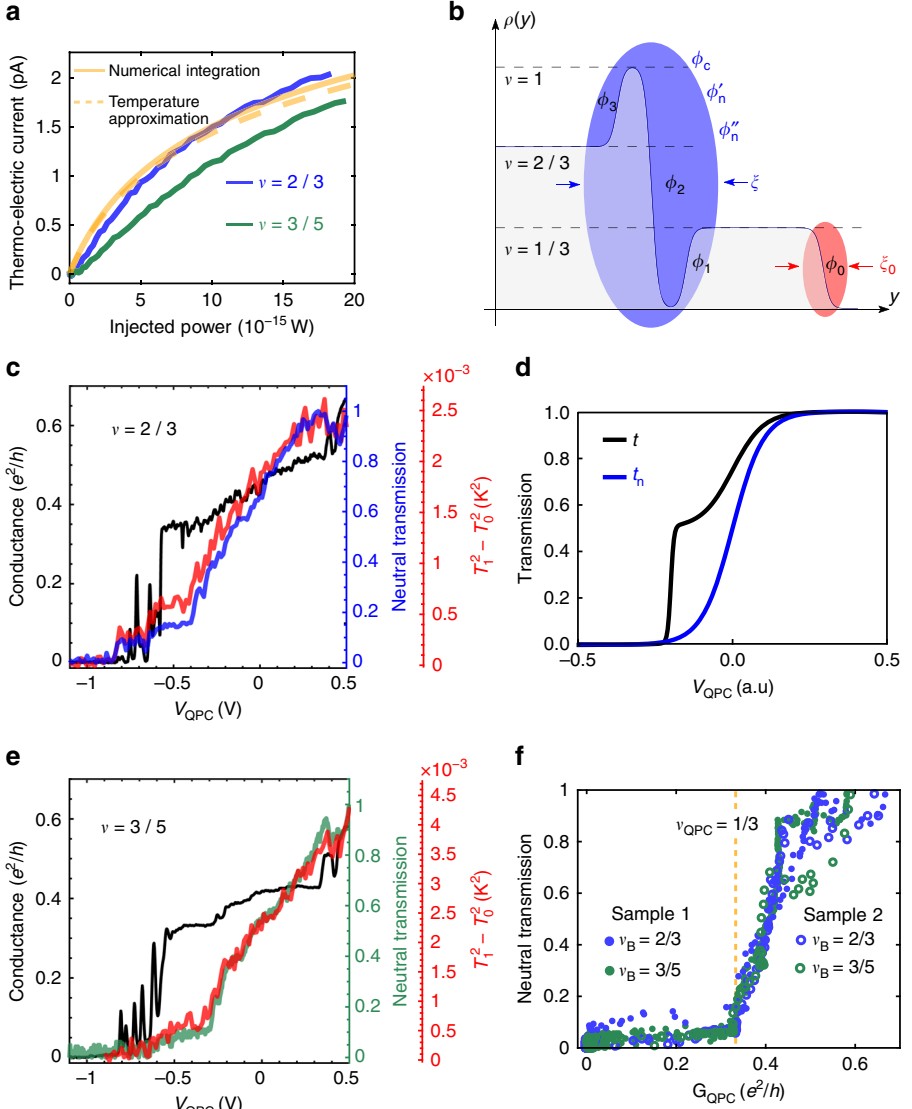

**Fig. 2** Transmission of neutral modes. **a** Evolution of the thermoelectric current as function of the Joule power applied on H at $\nu = 2/3$ (blue curve) and at $\nu = 3/5$ (green curve). Theoretical prediction of the thermoelectric current through the QD, orange dashed curve shows effective temperature approximation, orange solid curve shows result of exact numeric integration. **b** Theoretical model of the edge structure of the $\nu = 2/3$ state. Before equilibration four charge modes, $\phi_{0,1,2,3}$ are considered with respective velocity set by $\partial \rho(y)/\partial y$. After renormalization, the system consists of two downstream charge modes of different width $\xi$ and $\xi_0$ and two upstream neutral modes attached to the inner mode. **c** Black curve-left axis: Conductance at the QPC constriction as function of the QPC split gate voltage at $\nu = 2/3$, blue curve-right blue axis: Evolution of the neutral transmission as function of the QPC split gate voltage. Red curve-red axis: evolution of the temperature of the neutral modes, measured using excess noise at D1 as function of the QPC split gate voltage. **d** Theoretical charge and neutral transmission. **e** Similar to **c** at filling factor $\nu = 3/5$. **f** Neutral transmission as function of the conductance in the QPC constriction for the bulk filing factors $\nu_B = 2/3$ and $\nu_B = 3/5$. The plain and open symbols designate two different samples

(Fig. 1a), or the neutral modes excited by the hot spot at the back of contact H (Fig. 1c, e). In the latter case, two methods were employed to convert the energy carried by the neutral modes to a measurable charge current. The first utilized a quantum dot (QD, on the left) to convert a temperature gradient to a net thermo-electric current[20–24] as sketched in Fig. 1c. With the thermal distribution at the input of the QD being hotter than at the output, scanning the energy level of the QD leads to a net ther-moelectric current through the QD with an alternating polarity (Fig. 1d, g, and experimental details in Methods Section). The second approach used excess noise measurements, resulting from the upstream heat current impinging on a QPC (Fig. 1e)[11,25,26]. Here the input QPC of QD was tuned to transmission half while

the output-QPC was fully open; therefore turning the QD to a single QPC. The impinging neutral mode increases the electron temperature at the QPC and accordingly the Johnson–Nyquist noise measured at D1 (Fig. 1f).

The thermopower of a QD, subjected to such a thermal gradient, was studied both theoretically and experimentally[20–23]. On the theoretical side, one needs to obtain the thermoelectric current while considering a specific model of the edge modes structure (see below). On the experimental side, we can directly relate the thermoelectric current to the heat current carried by the transmitted neutral modes. Since half of the injected power into contact H, $P_H = 1/2 \times IV$, is dissipated on the hot spot at the back of the ohmic contact H, it is proportional to the heat

 

current carried away by the neutral modes, $P \propto (T_1^2 - T_0^2)$, where $T_1$ is the temperature of the hot neutral modes and $T_0$ is the temperture of the system (more details in Supplementary Note 4), this allows us to map the thermoelectric current to the heat flow (Fig. 2a). Utilizing this correspondence, the spatial distribution of the neutral modes can be ascertained by measuring their transmission through a QPC constriction. Applying constant power (~20 fW) to contact H and measuring the thermoelectric (TE) current across the QD as we gradually pinch the right-QPC (see details in Measurement technique), we find the evolution of heat current carried by the neutral modes ("neutral transmission"), that we present on Fig. 2c (blue curve) together with the conductance of the QPC (black curve). The neutral transmission was extracted by taking the corresponding power for a given measured TE current from the mapping in Fig. 2a (blue curve); normalized to a fully open right-QPC. A second measurement, using the noise thermometry technique described above was used to validate the neutral transmission. The results are presented on Fig. 2c (red curve), where we plot the evolution of $T_1^2 - T_0^2$ (which is proportional to the heat current carried by the neutral modes), where $T_1$ is measured and $T_0 = 30$ mK is the base fridge temperature. It is clear from Fig. 2c that these two methods led to consistent results. Comparing the neutral transmission with the charge transmission, one notices that ~80% of the neutral mode power is reflected when the inner charge mode, with conductance of $e^2/3h$, is reflected (at the beginning of the conductance plateau, around $V_{QPC} = -0.4$ V), showing that most of the upstream heat flow is "attached" to the inner charge mode. A very similar result was obtained for $\nu = 3/5$ with strong correlation between the reflection of the inner charge mode and the reflection of the heat modes (Fig. 2e).

**Theoretical model**. We now compare our experimental results to a theoretical model of the $\nu = 2/3$ state. The first model for such state was developed by MacDonald[15], that predicted two counter-propagating charged edge modes with respective conductance $e^2/h$ flowing downstream and $e^2/3h$ flowing upstream. Due to the absence of any experimental evidence of upstream current[27], Kane, Fisher, and Polchinski[16,28] introduced scattering between the above-mentioned channels leading to a single downstream charge mode with a conductance $2e^2/3h$ and an upstream neutral mode. Later, Meir et al.[17,18] refined this model for a soft-edge potential (which is the case for gate-defined edge, like presently) and proposed an edge modes picture presented on Fig. 2b. A smooth confining potential induces a non-trivial density variation near the edge: Starting from the bulk, the local filling factor goes from 2/3 to 1 creating an upstream charge mode of conductance $e^2/3h$. The subsequent filling factor drop goes from unity to 0 and therefore creates a charge mode going downstream with an associated conductance equal to $e^2/h$. Finally, an extra density hump reaching $\nu = 1/3$ creates a pair of $G = e^2/3h$ counter propagating channels. Taking into account, interactions and scattering between the channels leads to the presence of a decoupled channel with conductance $e^2/3h$ of width $\xi_0$ close to the edge, flowing next to an inner, wider channel of width $\xi$, also with conductance $e^2/3h$ accompanied by two neutral modes flowing upstream. Note, that there are several alternative pictures of QH states at filling factors 2/3 and 3/5 both with and without neutral upstream edge modes (see, e.g., ref. [29]). Moreover, edge reconstruction phenomena[30,31] (e.g., additional humps in the density) can affect both the microscopic and effective behavior of edge states. Therefore, we focus our discussion on the simplest effective model of edge states that is less sensitive to the details of

experimental situation. The theoretical model presented here utilizes the "Meir et al." edge structure where couplings between channels leads to different excitation. The outer edge mode is completely decoupled from all other channels, while the inner channel gives rise to excitations, each characterized by a charge $e^*$ and a scaling dimension $\Delta$ (see details in Supplementary Note 4), which allow us to calculate the TE current through a single energy level $\epsilon_0$ of the QD for this particular edge state picture and for dominant excitation with $e^* = 1/3$ and $\Delta = 1$:[32]

$$I_{TE} \propto [f_{out}(T_{out}, \epsilon_0)f_{in}(T_{in}, -\epsilon_0) - f_{in}(T_{in}, \epsilon_0)f_{out}(T_{out}, -\epsilon_0)] \tag{1}$$

where $f_{out}$, $f_{in}$, $T_{out}$, and $T_{in}$ are the occupation numbers and temperatures corresponding to the input and output side of the quantum dot. Considering that the two upstream modes originate from a hot reservoir at temperature $T_1$ and the downstream ones at temperature $T_0$, it is possible to express the QD's input/output occupation number as,

$$f_{in}(\epsilon_0) = \int dt \, e^{i\epsilon_0 t} \left[ \frac{T_0}{\sinh(\pi T_0(t - i\eta))} \right]^{\delta_0} \left[ \frac{T_1}{\sinh(\pi T_1(t - i\eta))} \right]^{\delta_1} \tag{2}$$

with $\delta_0 = 1/3$, $\delta_1 = 2/3$, $\Delta = \delta_0 + \delta_1 = 1$ and

$$f_{out}(\epsilon_0) = \int dt \, e^{i\epsilon_0 t} \frac{T_0}{\sinh(\pi T_0(t - i\eta))} = \frac{1}{e^{\epsilon_0/T_0} + 1} \tag{3}$$

Inserting Eqs. (2) and (3) in Eq. (1), one can calculate the evolution of the TE current in a QD with a single energy level for this particular edge modes picture (see Supplementary Note 4). These results, either using effective temperature approximation or exact numeric integration, are plotted in Fig. 2a. The good agreement between the theoretical and experimental results strengthen the validity of an edge picture that consists of two charge modes accompanied by two upstream neutral modes. Using the same theoretical picture, we have modeled the neutral and charge transmissions through the QPC constriction using a quasi-classical approximation[33]

$$t_i(V) = 1 / \left( 1 + e^{(V - V_i)/\delta V_i} \right) \tag{4}$$

where $i$ = inner, outer. In our model, there are two modes that are spatially separated; therefore, they will be centered around different gate voltage $V_i$ of the split gate QPC, and with different width corresponding to $\delta V_i$. The total charge transmission $t_{charge}(V) = (t_{inner}(V) + t_{outer}(V))/2$ is plotted in Fig. 2d together with the neutral transmission, $t_{neutral}(V) = t_{inner}(V)$, where both the neutral modes are located near the inner charge mode. As visible on Fig. 2d, this simple model is able to qualitatively explain the transmission of the neutral modes; and in particular, the observed vanishing of the neutral transmission when the conductance reaches $G_{QPC} = 1/3$. To compare in more details the measured transmission of the neutral modes in the two bulk fractions, $\nu_B = 2/3$ and $\nu_B = 3/5$, we have plotted each one as a function of the conductance of the QPC constriction $G_{QPC}$ (Fig. 2f). Surprisingly, they present a nearly identical behavior, which strongly suggests that, at least for these two hole-conjugate states, the transmission of the neutral modes is dictated by the conductance of the QPC and is poorly dependent on the bulk filling factor. We have plotted on the same figure (open symbols) the results of another similar sample (presented in Supplementary Note 1). It is clear that the curves present an overlap on a large $G_{QPC}$ region, which indicates that the structure of the neutral modes is universal and do not depend

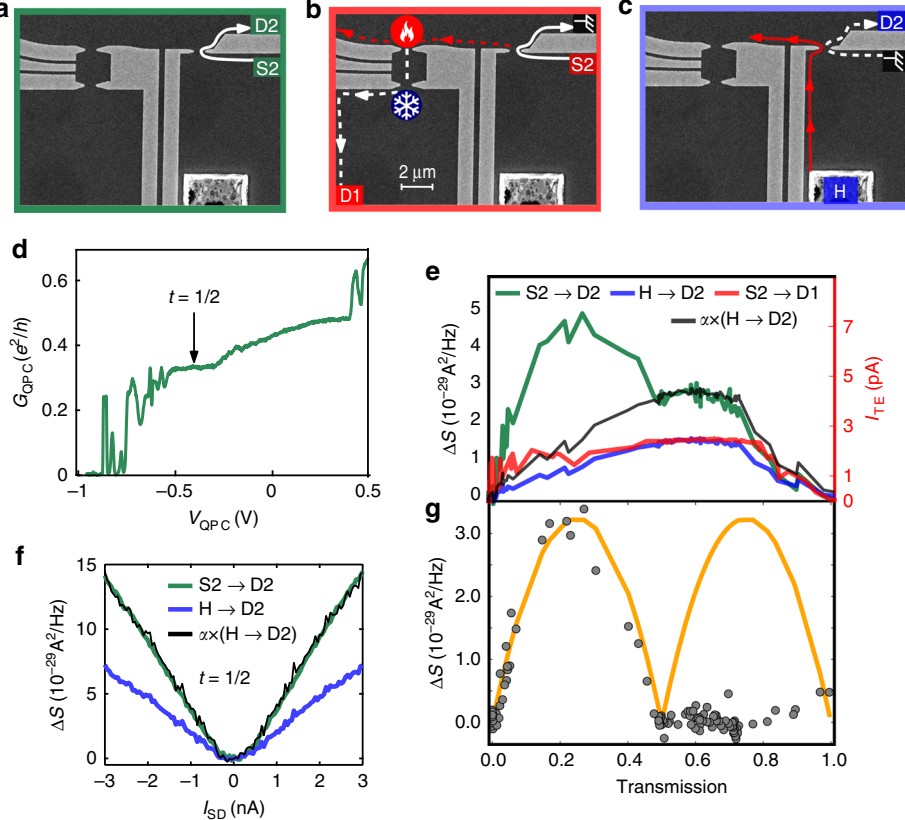

**Fig. 3** Noise measurements. **a** Sketch of noise measurement at D2 when sourcing charge current from S2 **b** Sketch of thermoelectric measurement at D1 when sourcing from S2, the impinging current generates heat carried upstream by the neutral modes and converted to thermoelectric current at the QD. **c** Sketch of noise measurement at D2 when sourcing from H, the upstream neutral modes excited at the hot spot increase the Johnson–Nyquist noise measured at D2 **d** Conductance of the QPC as function of the split gate voltage. **e** Green: Current fluctuations measured at D2 when sourcing from S2 as function of the transmission of the QPC. Blue: Current fluctuations measured at D2 when sourcing from H. Black: Same as blue multiplied by a scaling factor $\alpha = 2$. Red: Thermoelectric current measured at D1 when sourcing from S2. **f** Excess noise measured at D2 as function of the current at transmission half when sourcing from S2 (green) or H (blue). The black curve represents the noise from H to D2 (blue curve) multiplied by the scaling factor $\alpha = 2$. **g** Gray dots: Shot noise contribution $N_{SN} = N_{tot} - \alpha N_{th} = N_{S2 \to D2} - \alpha N_{H \to D2}$. Orange curve: expected shot noise from two separated ballistic charge channels

on the bulk state as well as the particular mobility or disorder in the QPC constriction.

**Excess noise subtraction.** Noise measurements reveal even more about energy exchange mechanisms between the charge and neutral modes. We first start from the configuration presented on Fig. 3a, where we measure excess noise in D2 when sourcing DC current from S2. As shown in Fig. 3e, f (green curves) and as reported previously[25,26,34,35], the noise remain finite when the QPC is tuned to the conductance plateau at $t_{QPC} = 1/2$ with $G_{QPC} = e^2/3h$ (Fig. 3d). This finite noise is even more puzzling since in recent experiment[19], it was shown that two spatially separated channels are present, excluding the presence of conventional shot noise at such transmission. Taking advantage of this device, we were able to show that a thermal contribution is added to the shot noise. In order to distinguish between the thermal noise and the partition noise of the quasiparticles (shot noise), the TE current in the QD was measured as function of the right QPC, but this time the current was sourced in S2 (Fig. 3b). The finite measured thermoelectric current presented in Fig. 3e (red curve) is a clear signature of heat dissipation occurring at the right-QPC constriction where the charge modes exchange energy with the neutral mode. Similarly, exciting neutral modes using contact H as shown in Fig. 3c, increases locally the electronic temperature at the

QPC—as discussed before (Fig. 3f, blue curve). Indeed, a similar dependence on the right QPC transmission, in both measurements, is observed in Fig. 3e. Subsequently, it is interesting to exclude the thermal contribution from the total excess noise measured at D2 when sourcing from S2 and be left with the shot noise contribution solely. In order to do so, we subtracted the thermal noise $N_{th}$ multiplied by a constant $\alpha$, that characterizes the neutral mode decay, from the total noise $N_{tot}$. The constant $\alpha$ was chosen such that the shot noise contribution at half transmission would be zero—as expected from a full transmission of one channel and a full reflection of the second. The result of this subtraction $N_{sn} = N_{tot} - \alpha N_{th}$ is plotted in Fig. 3g together with the expected shot noise (orange curve) for two charged channels giving a "double hump" shape (due to the $t(1-t)$ dependence of the noise of each channel). The shot noise $N_{sn}$ follows the expected behavior in the range 0–0.5 of total QPC transmission, but then collapses to 0 in all the region above half transmission. This would imply that the measured noise when the inner mode is being partitioned results solely from thermal fluctuations—consistent with the neutral modes being attached to the inner channel. This would furthermore indicate that, the outer mode can be partitioned like a ballistic channel while the inner one exhibits dissipation. More experimental and theoretical studies of this excess noise are required to have a full picture of such state.

## Discussion

Via studying the transmission of neutral modes through a QPC constriction at hole-conjugate states, $\nu = 2/3$ and $\nu = 3/5$, we showed that this structure is consistent with the presence of two upstream neutral modes attached to the downstream inner charge mode; being in agreement with the proposed model by Meir[17,18]. Moreover, this transmission of the neutral modes through a barrier formed by a QPC constriction is governed by its conductance and does not depend on the bulk filling factor. This suggests a universality of the neutral modes morphology, which should guide future theoretical models describing them and their interplay with the charge modes. More generally, these results mark a new step in the understanding of heat transport in the FQHE regime as well as for possible future implementation of controlled engineering of heat currents on nanoscale electronic devices.

## Methods

**Sample fabrication**. The samples were fabricated in GaAs–AlGaAs heterostructures, embedding a 2DEG, with an areal density of $(1.2–2.5) \times 10^{11}$ cm$^{-2}$ and a 4.2 K "dark" mobility $(3.9–5.1) \times 10^6$ cm$^2$ V$^{-1}$ s$^{-1}$, 70–116 nm below the surface. The different gates were defined with electron beam lithography, followed by the deposition of Ti/Au. Ohmic contacts were made from annealed Au/Ge/Ni. The sample was cooled to 30 mK in a dilution refrigerator.

**Measurement technique**. Conductance measurements were done by applying an a.c. signal with ~1 μV r.m.s. excitation around 1.3 MHz in the relevant source. The drain voltage was filtered using an LC resonant circuit and amplified by homemade voltage preamplifier (cooled to 1 K) followed by a room-temperature amplifier (NF SA-220F5).

For the thermoelectric current measurement, an alternating voltage $V_H$ at frequency $f \approx 650$ kHz was applied to contact H, giving rise to an upstream neutral modes with temperature proportional to $|V_H|$, producing a series of harmonics starting at $2f$. The heat reaching the upper side of the QD generates an alternating thermoelectric current flowing to drain D1. The signal is then filtered using an LC circuit with a resonant frequency at $2f = 1.3$ MHz amplified by a homemade voltage cold amplifier followed by a room temperature amplifier, and finally reaching a lock-in amplifier set to measure at frequency $2f$. The thermoelectric current was measured by sweeping the plunger gate for a given $V_H$ amplitude and then averaging the peak-to-peak thermoelectric current of more then 20 consecutive oscillations. The QD is in the metallic regime $\Delta E \ll k_B T \ll e^2/C$ where the temperature is above the level spacing $\Delta E$ and below the charging energy $e^2/k_B C = 290$ mK (see Coulomb diamond measurement in Supplementary Note 3).

**Data availability**. The data that support the plots within this paper and other findings of this study are available from the corresponding author upon request.

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

## Acknowledgements

A.R. and F.L. acknowledge Robert Whitney, Ady Stern, Yuval Gefen, Jinhong Park and Kyrylo Snizhko for fruitful discussions. M.H. acknowledges the partial support of the Minerva Foundation, grant no. 711752, the German Israeli Foundation (GIF), grant no. I-1241- 303.10/2014, the Israeli Science Foundation, grant no. ISF- 459/16, and the European Research Council under the European Community' s Seventh Framework Program (FP7/2007–2013)/ERC Grant agreement No. 339070.

## Author contributions

A.R., F.L., R.S., I.G., and M.H. designed the experiment, A.R., F.L., R.S., I.G., and D.B. preformed the measurements, A.R., F.L., R.S., I.G., D.B., and M.H. did the analysis, I.L.

developed the theoretical model. A.R., F.L., I.L., and M.H. wrote the paper. V.U. grew the 2DEG.

## Additional information

**Competing interests:** The authors declare no competing financial interests.

