## [Peer Review File · Nature Communications]

Reviewers' comments:

Reviewer #1 (Remarks to the Author):

The manuscript "Transmission of heat modes across a potential barrier" represents an impressive set of state-of-the-art experiments investigating the spatial distribution of the neutral heat modes in the edge states of the FQHE at $2/3$ and $3/5$ **.

It seems though that the interpretation of the results depend heavily on the theoretical model. In the abstract the $2/3$ and $3/5$ states are described as hole-conjugate states, i.e., $2/3 = 1-1/3$ and $3/5=1-2/5$. The presence of a neutral mode depend on $2/3$ and $3/5$ being hole-conjugate states. However, $2/3$ and $3/5$ can be alternatively understood using the composite fermion description. That is, $2/3$ consists of 2 filled composite fermion Landau levels with reverse flux attachment ($3/5$ is 3 filled composite fermion Landau levels). This alternative is not mentioned at all in the manuscript. Is there a compelling reason for this?

As I wrote at the beginning of this review, the experiments presented in this paper provide compelling evidence in support of the particular theoretical model of the edges at $2/3$ and $3/5$ presented in references 15 and 16. However, I believe that more attention should be paid to the possibility of alternative descriptions, perhaps without neutral modes. Is the data here totally inconsistent with a theory without neutral modes? Edge reconstruction in the FQHE is notoriously complicated and can depend on the nature of the confinement producing the edge. It seems theoretically possible that other edge reconstruction configurations could also be consistent with the presented data. Of course, one cannot expect the authors to exhaustively compare their results to all possible theories, however, a brief discussion of the complicated nature of these systems would improve the manuscript and make it scientifically more sound.

To conclude, with the additional discussions mentioned above I am inclined to find this manuscript acceptable for publication. These are remarkable experiments but the interpretation needs to be careful and tempered such that the data can stand on its own.

Below are a few technical comments on the manuscript itself:

- The QD should be labeled in Fig. 1 so there is no confusion.
- Starting on line 47 when discussing the structure of the FQHE states, are you discussing the edge in particular or the bulk. Please be more precise.
- In line 122, is there an obvious reason why the heat current is proportional the difference in squared temperature. I realize this is discussed in the supplemental, but if it is not obvious it should be noted it is discussed in the supplemental.
- Lines 150-153 are confusing. A "wider channel of with ϵ_0 "? From the figure ϵ_0 is drawn with a width that is narrower than ϵ . Also, why are you writing $e^2/3h$ and $1/3 \times e^2/h$? They both equal $e^2/3h$.
- Line 161: Why one level of the QD. There are many energy levels? Is it because the temperature is low so you only need the lowest level?
- Eq. 2 and 3. Where did f_U and f_D come from? Are they f_{in} and f_{out} ?
- Ady Stern's name is spelled wrong on line 316.

Reviewer #2 (Remarks to the Author):

The manuscript "Transmission of heat modes across a potential barrier" analyses heat transport across a quantum point contact in the fractional quantum Hall regime at filling factors $\nu=2/3$ and $\nu=3/5$. The thermoelectric effect across a quantum dot is used to convert heat currents into charge currents which can be measured subsequently. A theoretical modelling of the experimental findings is provided within the framework of bosonization which is a standard tool to describe

strongly interacting 1D systems.

The manuscript is well written and presents a nice and interesting piece of work that will certainly have a major impact on the mesoscopic transport community. The results are scientifically sound and convincing. The topic of this work is very timely as it provides both insights into (fractional) quantum Hall physics and strong interaction effects as well as into heat transport at the nanoscale which can be of relevance for future energy harvesting applications. I think that the results presented in this work have a number of important implications. First of all, they provide a first step towards a systematic investigation of heat transport in the quantum Hall regime. Furthermore, they provide new insight into the edge state structure of hole-conjugated fractional quantum Hall states. Finally, the manuscript may pave the way for future studies dealing with more complex setups like interferometers that can give even more insights into (fractional) quantum Hall physics.

It is for the above reasons that I believe that the manuscript definitely deserves publication in Nature Communications once the following questions and minor comments have been addressed by the authors.

Questions:

1. The manuscript explains in some detail the partitioning of the inner and outer charge mode at the quantum point contact as well as the behaviour of the neutral mode.

What about the partitioning of edge states at the QPCs that define the quantum dot? Do I understand correctly that the outer edge is transmitted with unit probability through the dot and the inner edge weakly coupled to the dot? If so, would no thermoelectric effect occur if the outer edge is weakly coupled to the dot (and the inner edge basically decouples from the dot)? Have the authors checked these different scenarios experimentally?

2. Fig. 2b) and Fig. S3: I could imagine that the paper becomes more accessible to a broad readership if a brief explanation of how the steps in the density are related to the edge states would be given.

Minor comments:

line 80: "Fig. 2c" should be replaced by "Fig. 1c"

Fig. 1 b) The caption mentions a high density of states region which should rather be called high occupation number region.

Eq. 3: T should be replaced by T_0

line 247: "red curve" should read "orange curve"

Response to Reviewers' comments:

Reviewer #1

> It seems though that the interpretation of the results depend heavily on the theoretical model.

Although the quantitative description of the experimental results points to a particular theoretical model, several qualitative results (such as, e.g., the presence of upstream heat flow and the absence of upstream charge currents) suggest the presence of upstream neutral modes. This more general result is independent of a particular theoretical model and agrees with previous experiments at filling factors $2/3$ and $3/5$.

> However, $2/3$ and $3/5$ can be alternatively understood using the composite fermion description.

[...]

> This alternative is not mentioned at all in the manuscript. Is there a compelling reason for this?

We limited the discussion to a single model of bulk states to avoid possible reader confusion, but in view of this comment we decided to add a short discussion of alternative descriptions (see also another comment below).

We thank the referee for this suggestion.

> Is the data here totally inconsistent with a theory without neutral modes?

We cannot claim the "total inconsistency", but we are not aware of any theoretical model (neither microscopic nor effective) without neutral upstream modes that would be compatible with current and previous experimental findings.

> Edge reconstruction in the FQHE is notoriously complicated and can depend on the nature of > the confinement producing the edge.

Indeed, even the model considered in the paper can be seen as an example of such complex reconstruction.

We have added a note that the edge structure strongly depends on the electrostatic potential profile at the edge.

> It seems theoretically possible that other edge reconstruction configurations could also be consistent with > the presented data.

Yes, there are several "layers" of theoretical description of the QH edge states. Different microscopic bulk models can produce the same effective low-energy projection in bulk, the same is true for the edge states. Finally, different effective bulk models can be consistent with the same effective edge model and vice-versa. Therefore, we focus our discussion on the simplest known to us effective model of edge states that agrees with the observations. Such model could be seen as a "common denominator" of possible microscopic edge/bulk pictures.

> ...however, a brief discussion of the complicated nature of these systems would improve the manuscript and > make it scientifically more sound.

This is a good suggestion. We added such discussion in the new version of the manuscript.

- The QD should be labeled in Fig. 1 so there is no confusion.

We labeled the QD on the figure. Let us know if you think it is clear enough.

- Starting on line 47 when discussing the structure of the FQHE states, are you discussing the edge in particular or the bulk. Please be more precise.

We changed the sentence to Two edge-model structures had been proposed for the most studied $\nu=2/3$ state

- In line 122, is there an obvious reason why the heat current is proportional to the squared temperature. I realize this is discussed in the supplemental, but if it is not obvious it should be noted it is discussed in the supplemental.

We added a mention to the supplementary

- Lines 150-153 are confusing. A "wider channel of width ϵ_0 "? From the figure ϵ_0 is drawn with a width that is narrower than ϵ . Also, why are you writing $e^2/3h$ and $1/3 \times e^2/h$? They both equal $e^2/3h$.

We indeed mixed up the positions of ξ and ξ_0 on the figure. We corrected it now. We also changed in the text the conductance of both channels to the same "style" for more clarity.

- Line 161: Why one level of the QD. There are many energy levels? Is it because the temperature is low so you only need the lowest level?

Indeed the quantum dot is in the metallic regime with a typical charging energy of 25 microVolts (times the elementary charge) that corresponds to a thermal energy of 290 mK (times the Boltzmann constant) which is an order of magnitude higher than the electronic temperature. The Coulomb diamond figure has been added to the supplementary materials as well as a mention in the methods.

- Eq. 2 and 3. Where did f_U and f_D come from? Are they f_{in} and f_{out} ?

We indeed changed the notation to f_{in} and f_{out} in all the text. There were no differences

- Ady Stern's name is spelled wrong on line 316.

Indeed I used a wrong transliteration for his name.

Reviewer #2 (Remarks to the Author):

Questions:

1. The manuscript explains in some detail the partitioning of the inner and outer charge mode at the quantum point contact as well as the behaviour of the neutral mode. What about the partitioning of edge states at the QPCs that define the quantum dot? Do I understand correctly that the outer edge is transmitted with unit probability through the dot and the inner edge weakly coupled to the dot? If so, would no thermoelectric effect occur if the outer edge is weakly coupled to the dot (and the inner edge basically decouples from the dot)? Have the authors checked these different scenarios experimentally?

The quantum dot is pinched so no full channel is going through it. It is actually very hard to experience Coulomb blockade physics with the inner channel at filling factor $2/3$ (due to the easy mixing between the channels :Nature Physics 13,491–496 (2017)). It would be indeed very interesting to be able to do the same experiment with the inner channel to see if the signal measured significantly differs.

2. Fig. 2b) and Fig. S3: I could imagine that the paper becomes more accessible to a broad readership if a brief explanation of how the steps in the density are related to the edge states would be given.

We added a short discussion about the link between density change and edge states.

Minor comments:

line 80: "Fig. 2c" should be replaced by "Fig. 1c"

Fig. 1 b) The caption mentions a high density of states region which should rather be called high occupation number region.

Eq. 3: T should be replaced by T_0

line 247: "red curve" should read "orange curve"

We corrected all these minor comments

REVIEWERS' COMMENTS:

Reviewer #1 (Remarks to the Author):

I appreciate the thorough answers to all my inquiries by the authors. I now fully recommend publication of the manuscript in its revised form in Nature Communications.

Reviewer #2 (Remarks to the Author):

The authors have addressed the questions and comments raised in the first round of review in a satisfying manner. Therefore, I recommend the manuscript for publication in Nature Communications.